# A Systematic Review of Genetic Polymorphisms Associated with Binge Eating Disorder

**DOI:** 10.3390/nu13030848

**Published:** 2021-03-05

**Authors:** Lucia Manfredi, Alessandra Accoto, Alessandro Couyoumdjian, David Conversi

**Affiliations:** Department of Psychology, University of Rome ‘La Sapienza’, 00185 Rome, Italy; manfredi.1654439@studenti.uniroma1.it (L.M.); alessandra.accoto@uniroma1.it (A.A.); couyoumdjian@uniroma1.it (A.C.)

**Keywords:** genetics, binge eating disorder, systematic review, obesity, polymorphisms, dopamine, serotonin

## Abstract

The genetic polymorphisms involved in the physiopathology of binge eating disorder (BED) are currently unclear. This systematic review aims to highlight and summarize the research on polymorphisms that is conducted in the BED. We looked for observational studies where there was a genetic comparison between adults with BED, in some cases also with obesity or overweight, and healthy controls or obesity/overweight without BED. Our protocol was written using PRISMA. It is registered at PROSPERO (identification: CRD42020198645). To identify potentially relevant documents, the following bibliographic databases were searched without a time limit, but until September 2020: PubMed, PsycINFO, Scopus, and Web of Science. In total, 21 articles were included in the qualitative analysis of the systematic review, as they met the eligibility criteria. Within the selected studies, 41 polymorphisms of 17 genes were assessed. Overall, this systematic review provides a list of potentially useful genetic polymorphisms involved in BED: 5-HTTLPR (5-HTT), Taq1A (ANKK1/DRD2), A118G (OPRM1), C957T (DRD2), rs2283265 (DRD2), Val158Met (COMT), rs6198 (GR), Val103Ile (MC4R), Ile251Leu (MC4R), rs6265 (BNDF), and Leu72Met (GHRL). It is important to emphasize that Taq1A is the polymorphism that showed, in two different research groups, the most significant association with BED. The remaining polymorphisms need further evidence to be confirmed.

## 1. Introduction

Binge eating disorder (BED) is a common and severe eating disorder characterized by recurrent episodes of eating large amounts of food without unhealthy compensatory measures (e.g., vomiting, excessive exercise), a feeling of loss of control during binge eating, and the experience of shame, distress, or guilt afterwards. The fifth edition of the *Diagnostic and Statistical Manual of Mental Disorders* (DSM-5) defined BED as at least one binge episode per week for three months [1]. A multisite study conducted in the United States found that 8% of people with obesity, and 20–30% of people with obesity who participate in weight loss programs also met the criteria for BED, suggesting that BED is more common in overweight and obese people or in clinical populations seeking treatment for weight loss [2,3,4]

The genetic polymorphisms involved in the pathophysiology of BED are currently unclear. The strongest known susceptibility locus for obesity is the fat mass and obesity-associated (FTO) gene [5]. Although the mechanisms by which FTO variants influence obesity are unclear, FTO associations with several eating disorders, including BED, are apparent. Indeed, variants of the FTO gene are associated with poor behavioral regulation and BED, suggesting a genetic role in the pathogenesis of this disorder [5].

Besides, evidence suggests that genes regulating serotonin (5-HT) and dopamine (DA) in the central nervous system (CNS) may play critical roles in the pathophysiology of BED [6,7,8]. Serotonin’s role in eating disorders has been extensively studied, but relatively few studies have focused on BED and even less on serotonergic genes’ role in developing this disorder [7]. There is some evidence that abnormalities in brain 5-HT may play a significant role in binge eating behavior. In particular, genes involved in the 5-HT transmission, such as the 5-HT transporter (5HTT) gene, may contribute to the biological susceptibility of BED [9]. There is also evidence that other serotonergic genes, 5-HT2C and 5-HT2A receptors, are involved in weight regulation and eating disorders [10,11,12].

The dopaminergic system plays a key role in the context of both impulsive and eating behaviors [13,14]. Alterations at dopamine D2 receptors (DRD2s) and their influence on eating behaviors have received considerable attention. Several studies using different experimental approaches have shown that dopaminergic signal transduction is crucial for diet-induced weight gain [6].

The ankyrin repeat and kinase domain containing 1 (ANKK1) gene is proximal to DRD2 and can affect its expression; they are biologically linked. Both the ANKK1 and DRD2 dopamine receptor genes have been significantly associated with addiction in several replicated studies [7]. Other studies have investigated the possible role of the dopamine active transporter (DAT) SLC6A3 and D3 receptors, which are localized in the limbic areas of the brain and are associated with cognitive, emotional, and endocrine functions. Unfortunately, there is not enough available information linking these genes to eating disorders [15]. 

The DAT is highly expressed in the human striatum (an integral part of the mesolimbic reward pathways), where it is a critical regulator of synaptic DA and the duration of DA activity [16].

An important regulator of dopaminergic neurotransmission is also the enzyme catechol-O-methyltransferase (COMT), which catalyzes the degradation of dopamine, particularly in the prefrontal regions of the brain [17].

The opioid system is also strongly involved in reward regulation and is known to promote eating behavior by enhancing the hedonic properties of palatable foods [7].

BDNF (Brain Derived Neurotrophic Factor) is a neurotrophic protein, a member of the nerve growth factor family, and is widely distributed in many regions of the brain, including the hippocampus, which is one of the major regulatory centers for eating behavior. BDNF controls several behaviors associated with the regulation of food intake by acting in both the appetite and satiety centers of the brain. In this competition, serum levels of BDNF were analyzed in several patients with eating disorders and were found to be significantly lower compared to healthy controls [12].

Food intake and body weight are regulated in the hypothalamus by the hormones leptin and ghrelin, neurotransmitters of the melanocortin system (e.g., agouti-related protein) and neuropeptide Y (NPY). The human prepro-NPY gene has been implicated in the regulation of a wide range of physiological central effects such as control of appetite and body weight homeostasis [18].

Ghrelin, a peptide hormone produced by the enteroendocrine cells of the gastrointestinal tract, is involved in hunger and food initiation. The action of ghrelin on dopamine neurons increases food motivation, so it seems reasonable to expect changes in ghrelin modulation of the mesolimbic system in obesity. Indeed, it has been suggested that the ability of ghrelin to increase the reward value of food may lead to overeating and thus obesity [8].

Fatty acid amide hydrolase (FAAH) is the major degrading enzyme of endocannabinoids. There is evidence linking this enzyme to obesity. Many studies found a significant association between FAAH and overweight/obese patients with BED [19].

It was found that in a sample of severely obese study participants, all those carrying gene mutations of the melanocortin 4 receptor (MC4R) gene met DSM-IV criteria for an eating disorder (BED). This finding suggests that MC4R variants represent a genetic susceptibility to BED [20].

In conclusion, the gene circadian locomotor output cycles kaput (CLOCK), which belongs to the positive regulatory branch of the system, may play a role in genetic susceptibility to obesity [21].

The aim of this systematic review, which to the best of our knowledge is not available in the literature, is to highlight and summarize the research on polymorphisms conducted in the BED. We looked for observational studies where there was a genetic comparison between adults with BED, in some cases also with obesity or overweight, and healthy controls (CTRs) or obesity/overweight without BED.

## 2. Materials and Methods

### 2.1. Protocol and Registration 

Our protocol was written using Preferred Reporting Items for Systematic Reviews and Meta-Analyses: The PRISMA Declaration (PRISMA) [22]. It is registered at PROSPERO International Prospective Register from Systematic Reviews (https://www.crd.york.ac.uk/prospero/ identification: CRD42020198645 (accessed on 22 August 2020)) 

### 2.2. Eligibility Criteria

The inclusion criteria were (1) articles published in English; (2) observational studies on possible polymorphisms in the BED group (with or without obesity or overweight); (3) adulthood; and (4) both sexes.

The exclusion criteria were (1) non-human (animal models); (2) non-BED group, but only clinical populations with other eating disorders, such as bulimia nervosa (BN) and anorexia nervosa (AN), or food dependency; (3) childhood, adolescence, or old age; and (4) reviews and meta-analysis.

### 2.3. Information Sources

To identify potentially relevant documents, the following bibliographic databases were searched without a time limit, but until September 2020: PubMed, PsycINFO, Scopus, and Web of Science. The final search results were exported to Mendeley, and duplicates were removed by the software.

### 2.4. Search

The following keyword string was used in all databases: (binge eating disorder) AND ((polymorphisms) OR (gene) OR (genes) OR (genotypes) OR (genetic)). The filters were Humans; English; Adult: >18 years. 

### 2.5. Study Selection

After the publications retrieved with our research strategy were uploaded to Mendeley, this review followed a two-step screening process to determine whether the articles met the inclusion criteria. In each step, two reviewers independently reviewed all identified publications. The publications agreed upon by both reviewers went to the next stage. If the two reviewers could not reach a consensus on the admissibility of the article, a third reviewer was called in to contribute to the final decision. 

As a first step, the evaluators had to assess the admissibility of the relevant articles based on the title and the summary. Articles (observational studies) that examined at least one polymorphism in BED (on topic and right population) were selected for the next step. This phase also included articles that required a complete reading of the article to check compatibility with the screening criteria, e.g., polymorphisms or samples that are not listed in the title or abstract. The others were excluded for the following reasons: (1) study of polymorphisms in another disorder (on topic and wrong population), e.g., polymorphisms related to bulimia; (2) no study of polymorphisms, but discussion of BED (not on topic and right population), e.g., treatment of BED; and (3) neither study of polymorphisms nor discussion of BED (not on topic and wrong population), e.g., treatment of bulimia. 

In the second step, both reviewers had to read the full text of all articles that had passed the first step to determine which articles should be included in the qualitative analysis according to the selection criteria.

### 2.6. Data Collection Process 

Two reviewers independently charted data from each eligible article. Any disagreements were resolved by a discussion between the two reviewers or further evaluation by a third reviewer. Data charting was implemented with the software Google Sheets. The collected data was entered into a spreadsheet that was available to the entire review team.

### 2.7. Data Items

We abstracted data on article characteristics (author(s) and year of publication), study populations (groups, number of subjects, sex, age, diagnostic criteria of BED), candidates polymorphisms, methodology (study design), and main findings.

### 2.8. Risk of Biases in Individual Studies

We assessed each study using the Newcastle–Ottawa quality rating scale (NOS) [23]. The NOS is a classic assessment tool that evaluates three aspects of studies: selection, comparability, and exposure. This scale has a score ranging from 0 to 9, and a study is considered to be of high quality if it achieves a score greater than 7. The calculation of the NOS scores was performed independently by two investigators, and any disagreement between the two investigators was resolved by discussion with a third investigator until a consensus was reached.

## 3. Results

### 3.1. Study Selection 

In total, 853 publications were identified in the initial research. After the removal of duplicates, a total of 601 articles were analyzed in the first phase of screening. Of these, 574 were excluded for the following reasons: on topic and wrong population, *n* = 93; not on topic and right population, *n* = 55; not on topic and wrong population, *n* = 426. The remaining 27 were analyzed in full text in the second phase of the screening process. Of these, 6 were excluded for the wrong sample: childhood/adolescence sample, *n* = 2; non-BED group, *n* = 4 (for further details, see Table A1). The remaining 21 were included in the qualitative analysis of the systematic review, as they met the eligibility criteria. See Figure 1 for a flowchart of the search process. 

### 3.2. Study Characteristics

In the systematic review, the considered articles were published between 1999 and 2020 in English. Across all the studies, the majority of patients is constituted by women. The age for case subjects (BED group, BED with obesity or overweight) ranged from 16 to 58 years, and for control subjects (normal-weight, obese or overweight without BED) ranged from 16 to 52 years. BED was assessed using primarily DSM-IV criteria. The study design is mainly case-control, except for two case-only articles. Within the selected studies, 41 polymorphisms of 17 genes were assessed. The characteristics of the 21 articles included in the systematic review are summarized in Table 1. 

### 3.3. Risk of Bias within Studies

The mean of the NOS score of the included studies was 6.62 (range 6–9), indicating that most studies were considered to be of high quality. Potential study biases resulted mainly from baseline characteristics between controls and cases and inappropriate selection of control groups (see Appendix A for further information). 

### 3.4. Results of Individual Studies 

#### 3.4.1. Serotonergic Genes

Five of the 21 selected articles investigate three polymorphisms of three serotonergic genes in BED: 5-HTT [9,24], 5-HT2C [10], and 5-HT2A [11,12]. 

Significant results were found by Monteleone et al., who analyzed the 5-HTTLPR polymorphism of the 5-HTT gene. Statistical analysis showed that both the genotype LL genotype and the L allele of 5-HTTLPR were significantly more frequent in obese people with BED [9]. Palmeira et al. also studied the same polymorphism and found no significant associations between 5-HTTLPR and BED [24].

As for the other serotonergic genes and their polymorphisms, no significant associations were found with BED. Burnet et al. considered the cys23ser polymorphism of the 5-HT2C gene in their study. The results showed that the genotype and allele frequencies were completely unchanged in BED and healthy control groups [10]. Both Ceccarini et al. and Ricca et al. investigated rs6311 (-1438A/G in the promoter region of the 5-HT2AR gene). Both studies found no significant association between this polymorphism and BED [11,12]. Besides, Ricca et al. found no significant differences between obese BED and non-obese BED individuals [11]. 

#### 3.4.2. Dopaminergic Genes

Nine studies examined the role of 19 polymorphisms of 6 dopaminergic genes in BED: DRD2 [15,24,25,26,27], ANKK1 27,28, OPRM1 [26], COMT [17,29], DAT1 [15,16], and DRD3 [15].

The Taq1A polymorphism of DRD2/ANKK1 genes was extensively analyzed in the selected articles, and these studies showed all the significant results mentioned above [15,24,25,26,27,28]. Davis et al. focused on Taq1A as a polymorphism of the DRD2 gene and found that BED subjects carrying the A1 allele of Taq1A showed higher reward sensitivity [25]. In contrast, Gonzalez et al. and Palmeira et al., who also analyzed Taq1A as a polymorphism of the DRD2 gene, did not find a significant result [15,24]. Davis et al. focused their analysis on DRD2 and OPRM1 because they are jointly associated with the function of reward mechanisms in the brain. They selected two polymorphisms of these two genes: Taq1A and A118G. Their results showed that a large proportion of the BED group with obesity carried the rare G allele. It turns out that in the genotype combination characterized by A1– and G+, 80% of the participants are BED. The opposite pattern holds for the genotype group characterized by A1+ and G–, where about 35% were obese binge eaters [26]. Davis et al. analyzed Taq1A as a polymorphism of the ANKK1 gene. In this study, they found that BED was significantly related to obesity with Taq1A (were in genotype group A2/A2) [27]. Palacios et al. also focused on Taq1A as a polymorphism of ANKK1 gene and found a significant association between Taq1A and BED with obesity [28].

Davis et al. analyzed the role of C957T polymorphism on the DRD2 gene and found that BED was significantly related to the C957T genotypes (were homozygous for the T-allele) [27], whereas Davis et al. also analyzed this polymorphism and found no significant relationship with BED [25,26].

Davis et al. also analyzed the polymorphisms rs2283265 and rs12364283 of DRD2. The only significant association found was that BED was less likely to carry the small T allele of rs2283265 with obese participants [27].

Davis et al. analyzed the −141C Ins/Del polymorphism on the DRD2 gene and found no significant association with BED [25,26,27]. 

Davis et al. also focused on three other polymorphisms of the DRD2 gene: −241 A/G, Taq1D and rs4648317. They found no significant association with BED [25]. 

Gervasini et al. and Leehr et al. analyzed Val158Met, which is a COMT polymorphism [17,29]. Leehr et al. found that within the group BED homozygous COMT Met/Met individuals with obesity showed stronger deficits in inhibitory control [17]. Gervasini et al. found no significant association with BED [29].

Palacios et al. analyzed seven other polymorphisms of the ANKK1 gene besides Taq1A: two polymorphisms in exon 2 (rs17115439 and rs4938013), one in exon 5 (rs7118900), one in exon 6 (rs11604671), and three in exon 8 (rs4938016, rs2734849, rs2734848). They did not find a significant connection between these seven polymorphisms and BED [28]. 

Regarding the DAT1 gene, both Davis et al. and Gonzalez et al. analyzed the 3-UTR-VNTR polymorphism [15,16]. Davis et al. found no significant group difference in DAT1 genotype frequency at baseline (the placebo condition) [16]. Gonzalez et al. also included the Ser9Gly (rs6280) polymorphism of the DRD3 gene in their study. They did not find a significant result for either polymorphism [15].

#### 3.4.3. Other Genes

Ten studies investigated the role of 19 polymorphisms of 8 other genes in BED: GR [30]; MC4R [20]; BDNF [12,24,31]; prepro-NPY [18]; prepro-GHRL [18,24,32]; FAAH [19]; FTO [5,24]; and CLOCK [21]. 

Cellini et al. analyzed four polymorphisms of the GR gene: exon 9-b (rs6198), ER22/23EK (rs6189–6190), N363S (rs56149945), and the intronic BclI (rs41423247). The results highlight two important findings. First, a significant correlation between the genotypes of BMI and GR was found for SNP rs56149945 (N363S) in obese patients and all ED groups, regardless of eating psychopathology. Besides, a significant association was found between the polymorphism rs6198 and the symptoms of BED [30].

Tortorella et al. performed a mutation scan in the group BED. They discovered two polymorphisms, Val103Ile and Ile251Leu, of the MC4R gene in BED in obese people. No variants of the MC4R gene were observed in the 40 controls [20].

Monteleone et al. analyzed the 196G/A polymorphism of BDNF and found no significant difference between patients with BED and healthy controls [31]. Ceccarini et al. analyzed rs6265 (196C/T) in the coding region (Val66Met) of BDNF and found that the Val66Met-SNP shows a strong association with BED patients [12]. Palmeira et al. analyzed two other polymorphisms of BDNF—rs16917237 and rs6265 (Val66Met)—and no significant associations were found between these two polymorphisms and BED [24]. 

Kindler et al. examined the Leu7Pro polymorphism of prepro-NPY gene, and the analysis of the diagnostic subgroup BED did not reveal any significant association [18].

Monteleone et al. examined two SNPs: G152A (Arg51Gln) and C214A (Leu72Met) of the GHRL gene. Their results showed that the Leu72Met was significantly more frequent in BED patients and was associated with a moderate but significant risk of developing an eating disorder. On the other hand, no significant difference in the distribution of Arg51Gln ghrelin gene variants was found between BED patients and healthy controls [32]. Kindler et al. analyzed Arg51Gln, Leu72Met, and Gln90Leu polymorphisms of the prepro-GHRL gene, and no significant relationship was found in the genotype distribution for any of the SNPs in BED patients [18]. Palmeira et al. also examined the polymorphisms rs696217 (Leu72met) and rs4684677(Gln90Leu) of the GHRL gene, and no significant associations were found between these two polymorphisms and BED [24].

Monteleone et al. analyzed cDNA 385C for a polymorphism of the FAAH gene. The result of the SNP did not correlate significantly with the presence of BED [19].

Both Cameron et al. and Palmeira et al. analyzed the rs9939609 polymorphism of the FTO and found no correlation between this polymorphism and BED [5,24]. Cameron et al. also examined four other FTO polymorphisms: rs9939609, rs8050136, rs3751812, rs1421085, and rs1121980. There were no significant differences in the frequencies of the FTO allele in BED with overweight or obesity [5]. 

In conclusion, Monteleone et al. examined an SNP, 3111T/C, from the CLOCK gene and genotype, and the allele frequencies did not differ significantly between controls and overweight/obese patients with and/or without BED [21].

## 4. Discussion

### 4.1. Summary of Evidence 

In the current systematic review, which investigated the genetic polymorphisms associated with BED, we found 11 polymorphisms of 9 genes that showed significant associations with BED: 5-HTTLPR (5-HTT), Taq1A (ANKK1/DRD2), A118G (OPRM1), C957T (DRD2), rs2283265 (DRD2), Val158Met (COMT), rs6198 (GR), Val103Ile (MC4R), Ile251Leu (MC4R), rs6265 (BNDF), and Leu72Met (GHRL). 

Regarding the serotonergic system, the only significant polymorphism was 5-HTTLPR (5-HTT). Monteleone et al. found that both the LL genotype and the L allele of 5-HTTLPR were significantly more frequent in BED than in healthy subjects [9]. Since the L allele of HTTLPR is associated with increased transcriptional efficiency, they hypothesized that subjects homozygous for this allele tend to express a higher number of 5HTT sites at their serotonergic synapses. This could theoretically imply higher 5HT reuptake activity, which could reduce 5HT availability in the synaptic cleft. Lower serotonergic tone could represent a vulnerability factor for binge eating behavior, as it has been demonstrated that a decrease in 5HT activity leads to compulsive or binge eating [33]. In summary, despite the small size of their sample, Monteleone et al. suggested that the 5-HTTLPR could have a role in the genetic susceptibility to BED [9]. It is worth noting that this, to our knowledge, is the only study finding a significant association between 5-HTTLPR and BED, so their hypothesis needs further evidence to be confirmed. It is essential to note that the Palmeira et al. study did not find an association between 5HTT polymorphism and BED. This critical difference could be explained, as Palmeira et al. did, by the sample size [24].

Some of the selected articles studied Taq1A as a polymorphism of DRD2 and others as a polymorphism of ANKK1. We found these two different approaches because, at first, Taq1A was thought to be located in the 3′-untranslated region of DRD2 [34]. However, some studies showed that this polymorphism does not reside in DRD2 but in an adjacent gene, ANKK1 [35,36]. 

Davis et al. found that Taq1A, as a polymorphism of the DRD2 gene, had a moderating influence on BED subjects’ relationship, who carried the A1 allele, and higher reward sensitivity. Given the evidence linking the A1 allele to reduced receptor density, they expected an inverse relationship between psychological measures of reward sensitivity and the presence of the A1 allele. One explanation for these results could be that the BED and obese subjects have a different genetic variant that interacts with the A1 allele to produce higher dopamine activity [25]. Despite the results of Davis et al. about the Taq1A on the DRD2 gene, the other two groups (Gonzalez et al. and Palmeira et al.) did not find this association. Both of them explained that no relevant genetic associations with BED could be due to the sample size [15,24,25].

Davis et al. found that 80% of BED participants with obesity had a genotype combination characterized by gain-gain of function genotype G+ and A1− (A118G and Taq1A polymorphisms of OPRM1 and DRD2 genes) [26]. Their findings suggest that individuals with BED are a specific subtype of obesity. They obtained evidence that biologically based hyperreactivity to the hedonic properties of food, coupled with motivation to engage in appetite-stimulating behaviors, may influence susceptibility to binge eating. This predisposition can be easily exploited in today’s environment with its unmissable and readily available overabundance of sweet and fatty things to eat [26]. 

The most consistent results of Davis et al. were seen with the A2/A2 genotype (A1− allele) of Taq1A, considered here as a polymorphism of the ANKK1 gene. In general, they found that this genotype had significantly higher levels of binge eating, hedonic eating, emotional eating, and food cravings in BED subjects with obesity compared to A1+ carriers, which are phenotypes to be expected in BED [27]. 

In line with Davis et al. studies, Palacios et al. found that Taq1A certainly has a crucial role in BED etiology in subjects with obesity [26,27,28].

Davis et al. found that the T/T genotype for C957T, a polymorphism of the DRD2 gene, showed higher scores only on the binge eating variable. Moreover, those who did not carry the rs2283265 T allele (viz. the G/G genotype) had significantly higher binge eating and emotional eating scores [27]. 

Leehr et al., investigating Val(108/158)Met, a polymorphism of COMT gene, found that Met/Met homozygous individuals with BED might represent a specific group in the BED spectrum, which shows higher behavioral impulsivity [17]. For the small size of the sample they used, this association must be interpreted with caution.

Overall, the described results offer initial support for the view that BED is a condition that may have its causal origins in a hypersensitivity to reward. Strong responsiveness to rewarding stimuli can considerably increase the likelihood of overconsumption, especially in our current environment with highly palatable food [27]. 

We also found other genetic polymorphisms related to GR, MC4R, BDNF, and GHRL genes in addition to the genetic polymorphisms involved in the serotonergic and dopaminergic systems. 

Cellini et al. looked at the distributions of genotypes and allele frequencies of the GR SNP rs6198 in all ED and obese groups and found a significant association exclusively with BED. Notably, BED patients showed the GG genotype. GG carriers (rs6198) showed higher eating and shape preoccupation compared to the AA and GA carriers [30]. The authors explained their findings by stating that cortisol secretion is an essential component of the stress response [37], and stress and negative affect are the most commonly cited antecedents of binge eating [38]. In addition, the increase in circulating GC concentrations enhances carbohydrate and fat intake [39]. It is essential to underline that the moderate association of the GG genotype of rs6198 in BED patients compared with the controls represents only a preliminary finding. This strict interpretation demands additional investigations to clarify these data.

Tortorella et al. found two polymorphisms, Val103Ile and Ile251Leu, of the MC4R gene in BED patients. This study suggests that variations in the involved gene do not represent a shared genetic vulnerability for BED [20]. These findings need further investigations better to comprehend the role of these two polymorphisms in BED.

Ceccarini et al. found that the rs6265 (196C/T), in the coding region Val66Met of BDNF gene, showed a strong association with BED patients. The authors underline that the association between Met/Met genotype and BED and AN is extremely interesting. These two conditions have different and specific phenotypic characteristics, suggesting that BDNF could have a crucial role in food intake control and participate in the regulation of both pathways [12]. It is important to note that a previous study [31] did not find this association. This difference could be due to the heterogeneity of the sample characteristics).

Monteleone et al. found a significant association between the Leu72Met polymorphism of the GHRL gene and BED. Their results seem to suggest that this ghrelin gene variant may confer a moderate but significant risk for developing BED, although they cannot exclude that some unexplored factors could be involved in this association [32]. Other studies [18,24] did not find this association between Leu72Met polymorphism of the GHRL gene and BED. As Palmeira et al. explained, the main reason could be the small sample size [24]. 

### 4.2. Limitations

Several limitations require to interpret the results of this review with caution:The number of studies exploring associations between many genetic polymorphisms and BED is limited, and in some cases, the sample size used was small.The relationship between some polymorphisms and BED could also be influenced by gene–gene or gene–environment interactions.We could not pool data collected for a meta-analysis due to the heterogeneity of the genetic polymorphisms observed.

### 4.3. Conclusions

This systematic review is the first to explore the presence of genetic polymorphisms associated with BED to the best of our knowledge. Overall, this systematic review provides a list of potentially useful genetic polymorphisms involved in BED: 5-HTTLPR (5-HTT), Taq1A (ANKK1/DRD2), A118G (OPRM1), C957T (DRD2), rs2283265 (DRD2), Val158Met (COMT), rs6198 (GR), Val103Ile (MC4R), Ile251Leu (MC4R), rs6265 (BNDF), and Leu72Met (GHRL).

It is important to emphasize that Taq1A is the polymorphism that showed the most significant association with BED in two different research groups. The remaining polymorphisms need further evidence to be confirmed.

## Figures and Tables

**Figure 1 nutrients-13-00848-f001:**
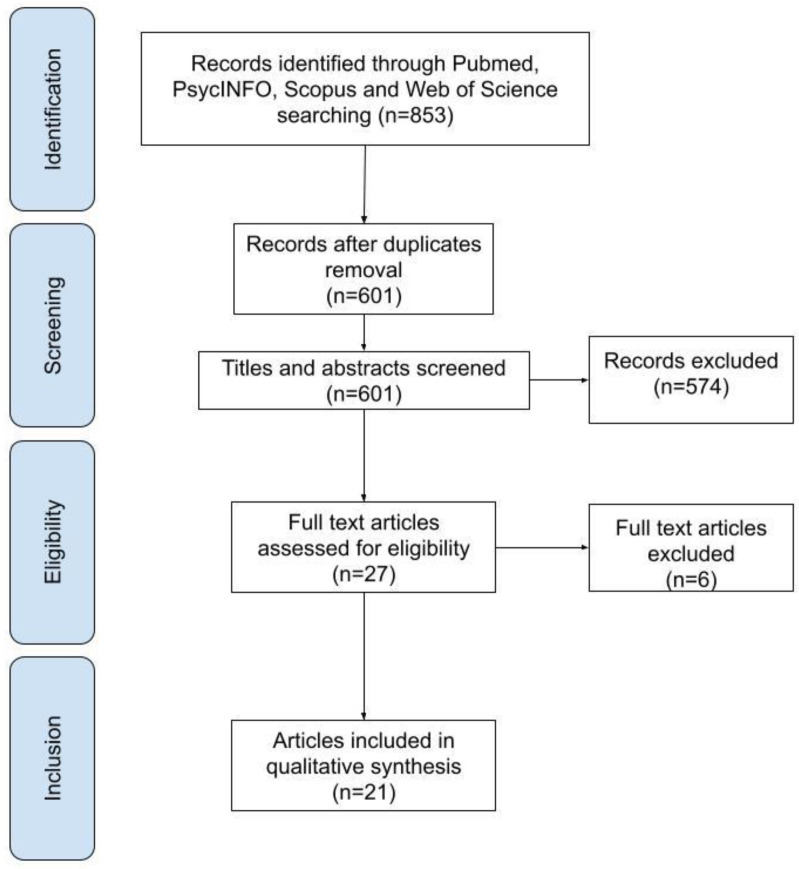
Flow diagram of selection process.

**Table 1 nutrients-13-00848-t001:** Characteristics of the included studies.

Article Characteristics	Study Populations			Candidates Genes and Polymorphisms Associated		Methodology	Results	NOS Score
Author(s) and (year)	Groups (N)	Sex: M, F; Age: years or Mean (sd)	Diagnostic criteria of BED	Gene(s)	Polymorphism(s)	Study design	Main findings	
Cameron et al. (2019) [5]	BED with overweight or obesity (73) Overweight or obese without BED (55) Normal weight (50)	F = 73; 44.2 (11.2) F = 55; 46.1 (11.9) F = 50; 43.8 (11.9)	DSM-IV	FTO (fat mass and obesity associated)	5 SNPs: rs9939609, rs8050136, rs3751812, rs1421085 and rs1121980.	Case-control	There were no significant between-group differences for frequencies of FTO allele, nor were there any significant anthropometric associations.	8
Monteleone et al. (2006) [9]	BED with obesity (77) CTRs (61)	F = 77; NA F = 61; NA	DSM-IV	5HTT (5HT transporter gene)	5HTTLPR	Case-control	Statistical analysis showed that both the LL genotype and the L allele of the 5HTTLPR were significantly more frequent in BED subjects. Moreover, the L allele was associated with a moderate but significant risk to develop BED.	6
Burnet et al. (1999) [10]	BN (40) BED (21) CTRs (92)	F = 40; 16–35 F = 21; 16–35 F = 92; age-matched	DSM-IV	5-HT2C (serotonin 2C receptor)	Serine (ser) is substituted for cysteine (cys) at codon 23 (cys23ser)	Case-control	Genotype and allele frequencies were entirely unaltered in both case groups, compared to screened healthy controls from the same population.	8
Ricca et al. (2002) [11]	BED (54) Obese non-BED (132)	NA; 39.5 (13.76) NA; 43.3 (12.75)	DSM-IV	5-HT2A (serotonin 2A receptor gene)	−1438 G/A	Case-control	No significant differences were found between obese BED and obese non-BED individuals, suggesting that this polymorphism does not genetically distinguish these two phenotypes.	6
Ceccarini et al. (2020) [12]	AN (311) BN (115) BE (130) CTRs (355)	M = 5, F = 306; 22.4 (± 8.9) M = 5, F = 110; 26.3 (± 9.4) M = 15, F = 115; 37.8 (± 18) M = 38, F = 317; 27.7 (± 7.9)	DSM-V	5-HT2AR (serotonin 2A receptor) BDNF (brain-derived neurotrophic factor)	1 SNP: rs6311 (−1438A/G) in the promoter region of 5-HT2AR 1 SNP: rs6265 (196C/T) in the coding region (Val66Met) of BDNF	Case-control	There was association between the rs6311 SNP of the 5-HT2AR gene and AN, but not with BN and BE. The Val66Met SNP in the coding region of BDNF showed a strong association with both AN and BE patients. Their data show that Met/Met genotype is present in 7.1% of AN, 6.1% of BN, and 9.2% of BE against only 2% of CTRs. 6	6
Gonzalez et al. (2019) [15]	AN (210) BN (80) BED (34)	F = 210; NA F = 80; NA F = 34; NA	DSM-5	DAT 1 (dopamine transporter) DRD2 (dopamine D2 receptor) DRD3 (dopamine D3 receptor)	1 SNP: VNTR (rs28363170) 1 SNP: Taq1A (rs1800497) 1 SNP: Ser9Gly (rs6280)	Case-only	There were no differences between AN, BN, and BED with regard to the distribution of the different genotypes.	3
Davis et al. (2007) [16]	BED (32) CTRs (46)	F = 84.4%; 33.7 (± 6.5) F = 97.8%; 32.9 (± 7.0)	DSM-IV	DAT1 (dopamine transporter)	3′-UTR VNTR	Case-Control	At baseline (the placebo condition), there was no significant group difference in DAT1 genotype frequencies. BED subjects with at least one copy of the 9-repeat allele showed a significant suppression of appetite in response to methylphenidate compared with controls with this allele, or to subjects with the 10/10 genotype (irrespective of diagnosis) whose drug response was indistinguishable from placebo.	7
Leehr et al. (2016) [17]	BED+ (BED with obesity) (21) OBED- (Obese controls without BED) (23) NWC (Normal-weight healthy controls) (25)	F = 21; 31.0 (12.3) F = 23; 31.7 (11.2) F = 25; 31.4 (10.9)	DSM-IV	COMT (catechol-O-methyltransferase)	Val(108/158)Met	Case-control	In the BED+ group, COMT Met/Met homozygous individuals showed stronger deficits in inhibitory control.	6
Kindler et al. (2011) [18]	AN (46) BN (30) BED (38) CTRs (164)	M = 1; F = 45; 30.4 M = 1; F = 29; 26.6 M = 1; F = 37; 39.3 F = 164; 32.2	DSM-IV	prepro-NPY gene (prepro neuropeptide Y) prepro-GHRL (preproghrelin)	1 SNP: Leu7Pro 3 SNPs: Arg51Gln, Leu72Met and Gln90Leu	Case-control	No difference was seen in the genotype distribution between patients with EDs and controls in any of the four SNPs. The analysis of the diagnostic subgroups AN, BN, and BED also did not reveal any significant differences compared with controls.	7
Monteleone et al. (2008) [19]	BED with obesity/ overweight (115) Obese without BED (74) Normal weight CTRs (110)	F = 115; 34.8 (11.1) F = 74; 37.5 (12.7) F = 110; 27.3 (6.8)	DSM-IV	FAAH (fatty acid amide hydrolase)	1 SNP: cDNA 385C to A	Case-control	Compared to healthy controls, the whole group of overweight/obese BED and non-BED patients had a significantly higher frequency of the CA genotype and the A allele of the FAAH gene cDNA 385C to A SNP. Moreover, the SNP resulted significantly correlated to the presence of overweight/obesity, but not to the occurrence of BED	7
Tortorella et al. (2005) [20]	BED with Obesity (48) BED without obesity (10) CTRs (40)	F = 48; 18–58 F = 10; 18–58 NA; 18–39	DSM-IV	MC4R (melanocortin-4 receptor gene)	mutational scanning	Case-control	Two polymorphisms, Val103Ile and Ile251Leu of the MC4R gene were detected in BED patients. No variants of the MC4R gene were observed in the 40 controls.	5
Monteleone et al. (2008) [21]	Overweight/obese patients wit3h and/or without BED (192) CTRs (92)	Obese with/without BED: M = 28; F = 164; NA M = 14; F = 79; NA	DSM-IV	CLOCK (Circadian locomotor output cycles kaput)	1 SNP: 3111T/C	Case-control	Genotype and allele frequencies did not significantly differ between normal-weight controls and overweight/obese patients with and/or without BED.	6
Palmeira et al. (2019) [24]	Overweight (24) Obese (69, of which 31 with BED) CTRs (62)	F = 24; 41.52 F = 69; 41.52 F = 62; 42.73	DSM-5	BDNF (brain-derived neurotrophic factor) GHRL (ghrelin gene) SLC6A4/5-HTT (serotonin transporter gene) DRD2 (dopamine D2 receptor) FTO (fat mass obesity-associate d gene)	2 SNPs: rs16917237 and rs6265 (Val66Met) 2 SNPs: rs696217 (Leu72met) and rs4684677(Gln9 0Leu) 5-HTTLPR 1 SNP: rs1800497 (Taq1A) 1 SNP: rs9939609	Case-Control	No significant associations were found between polymorphisms and BED. Of interest, a markedly lower frequency of the FTO rs9939609 obesity risk A allele was found in BED patients (0.290) in relation to the control group (0.402). Contrasting with anorexia nervosa and bulimia nervosa, their data suggest that rs9939609 A allele has no potential role in BED.	8
Davis et al. (2008) [25]	BED (56) Obese CTRs (51) Normal-weight CTRs (59)	M = 12, F = 44; 34.84 (6.41) M = 12; F = 39; 36.29 (6.34) M = 7; F = 52; 33.49 (7.53)	DSM-IV	DRD2 (dopamine D2 receptor)	6 SNPs: Taq1A, −141C Ins/Del, −241 A/G, Taq1D, C957T, and rs4648317	Case-Control	Among the six SNPs related to DRD2, the findings for the Taq1A are perhaps of greatest interest given its considerable links to addictions in general, and obesity more specifically. BED and obese participants reported greater reward sensitivity compared to those with normal weight. The Taq1A genotype had a moderating influence on this relationship. Higher reward sensitivity was only observed in BED and obese subjects who carried the A1 allele.	8
Davis et al. (2009) [26]	BED with obesity (66) Obese CTRs (70)	M = 13, F = 53; 34.7 (6.5) M = 17, F = 53 37.0 (6.7)	DSM-IV	DRD2 (dopamine D2 receptor) OPRM1 (mu-opioid receptor)	3 SNPs: taq1A, −141c Ins/del, C957T 1 SNP: A118G	Case-Control	Their results show that a larger proportion of the obese control group carried the rare A1 allele whereas for the latter comparison, a greater proportion of the BED group carried the rare G allele. It can be seen that in the genotype combination characterized by A1− and G +, 80% are BED participants whereas only 20% are obese controls. The opposite pattern pertains for the genotype group carrying A1+ and G−, where about 65% were obese controls and about 35% were obese binge eaters.	8
Davis et al. (2012) [27]	BED with obesity (79) Obese without BED (151)	M = 12, F = 67; 38.6 (7.2) M = 47, F = 104; 38.7 (7.1)	DSM-IV	DRD2 (dopamine D2 receptor) ANKK1 (D2 receptor gene)	4 SNPs: rs1799732 (−141C ins/del), rs6277 (C957T), rs2283265 and rs12364283 1 SNP: rs1800497 (Taq1A)	Case-Control	Compared to weight-matched controls, BED was significantly related to the rs1800497 (were in the A2/A2 genotype group) and rs6277 (were homozygous for the T allele) genotypes that reflect enhanced dopamine neurotransmission. BED participants were also less likely to carry the minor T allele of rs2283265.	8
Palacios et al. (2018) [28]	Obese with BED (25) Obese without BED (25) CTRs (100)	M = 9; F = 16; 27–53 M = 12, F = 13; 30–52 M = 38; F = 62; 21–38	DSM-5	ANKK1	8 SNPs: Two polymorphisms in exon 2 (rs17115439 and rs4938013), one in exon 5 (rs7118900), one in exon 6 (rs11604671), and four in exon 8 (rs4938016, rs2734849, rs2734848 and rs1800497).	Case-control	After ANKK1 sequencing we did not find any mutations; however, rs1800497 (also known as Taq1A) in exon 8, showed an association with BED. Remarkably, for this study, the number of individuals for this polymorphism and additive model was sufficient to derive strong statistical power.	8
Gervasini et al. (2018) [29]	AN (199) BN (74) BED (30)	F = 199; 16.94 (± 4.58) F = 74; 17.76 (± 4.98) F = 30; 20.18 (± 7.38)	DSM-5	COMT (catechol-O-methyltransferase)	1 SNP: Val158Met	Case-only	The distribution of the different genotypes was similar in the three diagnosis groups and was comparable to the 1000 genomes database.	3
Cellini et al. (2010) [30]	AN (118) BN (108) BED (62) Obese non-BED (177) CTRs (107)	F = 94.4%; 19.21 (± 3.05) F = 98.2%; 20.64 (± 6.17) F = 87.3%; 23.50 (± 8.69) F = 80.2%; 24.00 (± 7.07) NA; 27.3 (± 1.5)	DSM-IV	GR (Glucocorticoids receptor)	4 SNPs: exon 9-b (rs6198), ER22/23EK (rs6189–6190), N363S (rs56149945) and the intronic BclI (rs41423247)	Case-Control	The results highlight two major findings. First, a significant association between BMI and GR genotypes was identified for the SNP rs56149945 (N363S) in obese patients and in all the ED groups, independent of eating psychopathology. Moreover, a significant association between the rs6198 polymorphism and the binge eating symptoms was detected.	7
Monteleone et al. (2006) [31]	BED (84) BN (126) CTRs (121)	F= 84; NA F= 126; NA F= 121; NA	DSM-IV	BDNF (brain-derived neurotrophic factor)	1 SNP: 196 G/A	Case-control	No significant differences were found in the frequencies of the 196G/A variants of the BDNF gene among patients with BN or BED and healthy controls.	6
Monteleone et al. (2007) [32]	BED (90) CTRs (119)	F = 90; NA F = 119; NA	DSM-IV	GHRL (ghrelin gene)	2 SNP: G152A (Arg51Gln) and C214A (Leu72Met)	Case-control	Statistical analyses showed that the Leu72Met ghrelin gene variant was significantly more frequent in BED patients and was associated with a moderate, but significant risk to develop binge eating disorder. No significant difference, instead, emerged in the distribution of Arg51Gln ghrelin gene variant between BED patients and healthy controls.	8

NOS: Newcastle–Ottawa scale; N: number; M: male; F: female; sd: standard deviation; BED: binge eating disorder; BN: bulimia nervosa; CTRs: controls; SNPs: single nucleotide polymorphisms; AN: anorexia nervosa; BE: binge eating; NA: not available; DSM-V: *Diagnostic and Statistical Manual of Mental Disorders*, 5th edition; DSM-IV: *Diagnostic and Statistical Manual of Mental Disorders*, 4th edition.

## Data Availability

Data are contained within the article or Appendix A.

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
