# Peer review of "A Systematic Review of Genetic Polymorphisms Associated with Binge Eating Disorder"

_nutrients, 2021, doi:10.3390/nu13030848_

Round 1

Reviewer 1 Report

The present study is a systematic review about the genetic polymorphsims associated to BED. It includes the most important observational studies that have been studied in the field to date and it is the firts study reviewing these evidences. The methodology seems appropiate although it would be desirable a better description of the included studies according to Newcastle-Otawa quality rating scale. It would be also interesting include data about the selected studies ( type of study , alhough all of them are observational studies, sample size, polimorphism studied and methodology implemented for that, main results... ) to follow better the  results section. The discussion should be more integrative and include explanations about the differences found between studies. For instance, It is not included  explanations about the differents results in Monteleone and Palmeira studies.

Reviewer 2 Report

The authors conducted a systematic review of the genetic polymorphisms associated with binge eating disorder (BED). This work is of great importance as the genetic factor included in the pathophysiology of BED is less clear. Here are some comments, which may improve the manuscript a bit.

  1. Line 33-34. It would be helpful to clearly state which population is the descriptive numbers (8% and 20-30%) apply to, is it globally, or a specific country, or population?
  2. Use the full name of DAT or DAT1 when it first appeared in the context.
  3. The introduction part (beginning from the second paragraph) is a list of genetic factors and biomarkers that are potentially associated with eating disorders. While this information is useful, the manuscript would be improved by reorganizing the structure. This information would be more appropriate for the Discussion part.
  4. Table S1 would be helpful if included in the main text, rather than a supplementary document, if space allows.
